# Autonomous Defense Based on Biogenic Nanoparticle Formation in Daunomycin-Producing *Streptomyces*

**DOI:** 10.3390/microorganisms13010107

**Published:** 2025-01-08

**Authors:** Karel Beneš, Vladislav Čurn, Baveesh Pudhuvai, Jaroslav Motis, Zuzana Michalcová, Andrea Bohatá, Jana Lencová, Jan Bárta, Michael Rost, Andreas Vilcinskas, Vladimír Maťha

**Affiliations:** 1VUAB Pharma A.S, Nemanicka 2722, 370 01 České Budějovice, Czech Republic; kbenes@vuab.cz (K.B.); jmotis@vuab.cz (J.M.); zmichalcova@vuab.cz (Z.M.); vladimir.matha@seznam.cz (V.M.); 2Department of Genetics and Biotechnology, Faculty of Agriculture and Technology, University of South Bohemia in České Budějovice, Studentská 1668, 370 05 České Budějovice, Czech Republic; rost@fzt.jcu.cz; 3Department of Plant Production, Faculty of Agriculture and Technology, University of South Bohemia in České Budějovice, Studentská 1668, 370 05 České Budějovice, Czech Republic; pudhuvaibaveesh@gmail.com (B.P.); bohata@fzt.jcu.cz (A.B.); lencova@fzt.jcu.cz (J.L.); barta@fzt.jcu.cz (J.B.); 4Branch Bioresources, Fraunhofer Institute for Molecular Biology and Applied Ecology (IME), Ohlebergsweg 12, 35392 Giessen, Germany; andreas.vilcinskas@ime.fraunhofer.de; 5Institute for Insect Biotechnology, Justus-Liebig-University Giessen, Heinrich-Buff-Ring 26-32, 35392 Giessen, Germany

**Keywords:** *Streptomyces coeruleorubidus*, anthracyclines, medium optimization, production strain development, iron chelators, vivianite, daunomycin-iron organic complex

## Abstract

Daunomycin is a chemotherapeutic agent widely used for the treatment of leukemia, but its toxicity toward healthy dividing cells limits its clinical use and its production by fermentation. Herein, we describe the development of a specialized cultivation medium for daunomycin production, including a shift to oil rather than sugar as the primary carbon source. This achieved an almost threefold increase in daunomycin yields, reaching 5.5–6.0 g/L. Daunomycin produced in the oil-based medium was predominantly found in the solid sediment, whereas that produced in the sugar-based medium was mostly soluble. The oil-based medium thus induces an autonomous daunomycin-resistance mechanism involving biogenic nanoparticle formation. The characterization of the nanoparticles confirmed the incorporation of iron and daunomycin, indicating that this approach has the potential to mitigate cytotoxicity while improving yields. The presence of proteins associated with iron homeostasis and oxidative stress responses revealed the ability of the production strain to adapt to high iron concentrations. Our findings provide insight into the mechanisms of biogenic nanoparticle formation and the optimization of cultivation processes. Further investigation will help to refine microbial production systems for daunomycin and also broaden the application of similar strategies for the synthesis of other therapeutically important compounds.

## 1. Introduction

Daunomycin (also known as daunorubicin) is an anthracycline antibiotic originally isolated from the bacterium *Streptomyces peucetius* that is used for the treatment of various cancers, particularly leukemia. Its therapeutic efficacy primarily reflects its ability to intercalate into DNA, disrupting replication and transcription in rapidly dividing cells. Daunomycin also serves as a precursor for the synthesis of more advanced anthracyclines [1]. However, its clinical use is compromised by its inherent cytotoxicity [2] and this also affects the microbial strains used for its production, posing significant challenges for industrial-scale synthesis [3,4].

The biological activity of daunomycin is dependent on its interaction with iron, particularly via the Fenton reaction. Iron catalyzes the production of reactive oxygen species (ROS) from hydrogen peroxide, inducing oxidative stress that leads to cellular damage [5]. Although iron is needed for numerous physiological functions, excess concentrations exacerbate oxidative stress. The therapeutic use of iron chelators to mitigate daunomycin toxicity [6] underscores the need for more information about the interplay between these two components and their influence on the pharmacological properties and therapeutic efficacy of daunomycin and the efficiency of production by fermentation. Research has focused on the toxicological effects of daunomycin in eukaryotic models, whereas the effects on prokaryotic cells are not understood in detail [7]. The biosynthetic gene clusters responsible for the biosynthesis of polyketide and sugarmoieties in daunomycin also include the regulatory genes and transcriptional repressor, which promotes the transcriptional control, self-resistance, and feedback regulation of the entire synthesis pathway [8]. Most studies have attempted to enhance the biosynthesis of daunomycin by inducing resistance, particularly through the genetic modification of efflux pumps in the production strain [9,10].

In order to increase metabolite production, we focused on the physiological adaptation of the strain to its toxic metabolites. We developed a cultivation medium specifically designed to promote auto-resistance based on autonomous defense via biogenic nanoparticle formation (ADBN). To enhance auto-resistance to the strain’s own toxic metabolites, we leveraged the fundamental properties of daunomycin, particularly its hydrophobicity and iron-chelating ability. Our strategy combined olive pomace oil in the medium and the bacteria’s propensity to form iron nanoparticles as carriers for the transport and inactivation of daunomycin [10]. We also investigated the effects of daunomycin on eukaryotic and prokaryotic cells (especially the bacteria involved in the production of daunomycin), a dual approach that will enhance our understanding of daunomycin’s mechanisms of action and potential applications in both eukaryotic and prokaryotic systems.

Olive pomace oil is a source of both lipids and polyphenols [11,12]. The unique amphiphilic nature of the lipids enables them to form a protective shell around the nanoparticles, where the hydrophilic heads interact with surrounding water molecules and the hydrophobic tails point towards the nanoparticle core [13]. This leads to the formation of a stable bilayer, which prevents nanoparticle aggregation [14]. The lipids in the medium thus control the growth and morphology of the nanoparticles [15]. The polyphenols enhance nanoparticle synthesis by initiating redox reactions when mixed with metal ions, reducing them to their metallic state [16]. The versatile nature of polyphenols also allows them to influence the size, shape, and characteristics of nanoparticles [17,18], and to confer colloidal stability by forming a protective layer that prevents agglomeration [19,20].

*Streptomyces* spp. can act as biofactories for the conversion of metal ions into nanoparticles [21], offering precise control over size and shape while ensuring stability, making this biological synthesis route a preferred choice [22,23]. By systematically optimizing iron levels, we facilitated the formation of daunomycin–iron complexes that reduce the solubility and bioavailability of daunomycin. Rather than sequentially testing different parameters, we approached optimization as a holistic challenge, thus accounting for interdependencies among key variables to improve performance and efficiency. The multifactorial and empirical nature of our optimization process—assessed by measuring daunomycin yields—highlights its intricacy and reinforces our objective to maximize production. Our findings not only validate our initial hypotheses but also offer valuable insights for future research aiming to optimize production processes in this field.

## 2. Materials and Methods

### 2.1. Chemicals

Defatted soy grits were supplied by MP Biochemicals (Irvine, CA, USA). Potato starch, G-glucose, yeast extract, NaCl, MgSO_4_, FeSO_4_, Fe_2_(SO_4_)_3_, glycerol, organic solvents and buffer salts were provided by Avantor (Randor, PA, USA). Dried baker’s yeast (Instaferm) was provided by Lallemand (Madrid, Spain), and brewer’s yeast was obtained from a local brewery (Budvar, České Budějovice, Czech Republic). Hemp oil and rapeseed oil were provided by a local supplier, and extra virgin and pomace olive oils were provided by Ondoliva (Navarra, Spain).

### 2.2. Bacterial Strain and Isolation

The original bacterial strain (*Streptomyces coeruleorubidus*) was isolated from mosquito larvae collected from the Vltava River pool in spring 2019. Early identification was confirmed by morphological assessments, phenotypic characterization, and rDNA sequencing. The strain is deposited in the Retorta s.r.o. strain collection under code VR 2019.

### 2.3. Triclosan-Induced Mutation

To enhance the production of daunomycin, triclosan was used to induce mutations in *S. coeruleorubidus* due to its known effect on enoyl-acyl carrier protein reductase, a key enzyme involved in fatty acid synthesis [24]. Solid potato dextrose agar (PDA) medium was mixed aseptically with triclosan at final concentrations of 0.1, 0.5, 1, and 10 µM. Subcultures were maintained for 10 passages to stabilize the mutant strains, which were tested for increased productivity on standard and modified media.

### 2.4. Olive Pomace Oil-Resistant Strain Development

The production strains were exposed to UV light at a dose of ~10 W/m^2^ for ~30 min as previously described [25,26] before cultivation on solid or in liquid medium containing 10% olive pomace oil as the sole carbon source.

### 2.5. Protoplast Formation

Protoplasts of *S. coeruleorubidus* were prepared as previously described [27,28] with modifications. Briefly, spores were inoculated in yeast extract malt extract (YEME) medium supplemented with 0.5% glycine and 5 mM MgCl_2_, followed by incubation at 28 °C, shaking at 220 rpm, for 48 h. Protoplasts were generated from the triclosan-resistant and olive pomace oil-adapted strains. The cultures were centrifuged at 4000 rpm for 10 min and washed with 10.3% sucrose. The cell pellet was washed with Medium P [27] then resuspended in 2 mg/mL lysozyme to remove the cell wall. The resulting protoplasts were washed and resuspended in 40% PEG 4000 to induce fusion and regeneration [29]. Colonies were selected by observing the darkening of the surrounding iron-containing agar.

### 2.6. Working Bank Preparation

Surface mycelia in a Petri dish were overlaid with 5 mL of 25% glycerol and scrubbed to create a cell suspension, which was processed in a PTFE-glass homogenizer for 15–20 min until the liquid was homogeneous. We then prepared 600 µL aliquots in Eppendorf tubes or cryo-tubes and stored them at −20 °C. Contamination was checked after 3 days by inoculating PDA with 5 µL of the cell suspension.

### 2.7. Inoculum Preparation

The fermentation inoculum was prepared from the working bank by transferring 2.5 mL of the working bank solution into 50 mL cultivation medium in a 250 mL Erlenmeyer flask. The culture was incubated at 28 °C, shaking at 220 rpm (orbit 25 mm). The culture was deemed ready for inoculation after 36–40 h (±2 h) in basal medium composed of 30 g/L soy flour, 5 g/L starch, 5 g/L glucose, 5 g/L yeast extract, 1 g/L K_2_HPO_4_, and 1 g/L NaCl. The pH was adjusted to 7.0 using 2 M NaOH or 2 M HCl as required.

### 2.8. Production Medium

We added 2.5 mL of the vegetative culture to 50 mL production medium in a 250 mL Erlenmeyer flask (5% inoculum). The culture was incubated at 28 °C, shaking at 220 rpm. The production medium was composed of 10 g/L baker’s yeast, 20 g/L soy flour, 100 g/L pomace olive oil, 5 g/L yeast extract, 5 g/L glycerol, 2 g/K K_2_HPO_4_, 1 g/L MgSO_4_.7H_2_O, 3 g/L CaCO_3_, and 3.6 g/L FeSO_4_.7H_2_O. The pH was adjusted to 5.9–6.1 using 2 M NaOH or 2 M HCl as required.

### 2.9. Effect of Carbon and Nitrogen Sources on Daunomycin Production

The influence of different carbon sources on daunomycin production was evaluated by replacing starch in the production medium with alternatives such as soybean oil, refined rapeseed oil, crude rapeseed oil, virgin olive oil, pomace olive oil, mineral oil, and crude hemp oil. Similarly, the impact of different nitrogen sources was assessed by replacing soy flour in the production medium with casein, yeast extract, defatted soy flour, soy grits, peptones, (NH_4_)_2_SO_4_, and KNO_3_.

### 2.10. Effect of Iron on Daunomycin Production

The medium containing optimized carbon and nitrogen sources was supplemented with iron sources such as 2.1 g/L FeCl_3_, 3.6–7.2 g/L FeSO_4_ and 5.2–10 g/L Fe_2_(SO_4_)_3_, corresponding to final concentrations of up to 1.44 g/L Fe^2+^ and 2.79 g/L Fe^3+^.

### 2.11. Direct Detection of Iron

Samples for scanning electron microscopy (SEM) and energy dispersive X-ray spectroscopy (EDS) were mounted on conductive stubs and a thin conductive coating was applied if necessary. A JSM-7401F instrument (Jeol, Tokyo, Japan) was calibrated and the accelerating voltage was set to 10–20 kV. Secondary electron imaging was used to visualize the samples and identify areas of interest. The EDS detector was activated to capture X-ray emissions from the samples and the resulting spectrum was analyzed for characteristic iron peaks, particularly the Fe Kα peak at ~6.4 keV. Elemental mapping was used to assess spatial distribution. The detected iron was quantified using SEM-EDS software (Ver. XM2), and the results were validated with standard reference materials.

### 2.12. X-Ray Diffraction of Iron-Containing Samples

The iron-containing samples were analyzed by X-ray diffraction (XRD) using an X’Pert PRO XRD system (PANanalytical, Almelo, The Netherlands) equipped with a sample spinner, goniometer, and 1D XCelerator detector. The radiation source was an X-ray tube with cobalt anode materials. The analysis parameters were set as follows: generator −40 mA, 35 kV, measurement temperature −25 °C, PSD mode—scanning, PSD length—2.12 °2Th., offset—0.00 °2Th., divergence slit type—fixed, divergence slit size—1.00°.

### 2.13. Monitoring of Daunomycin Production

Culture samples were collected after 120, 168, 216, and 264 h, with the possibility of extending to 312 h. For each sampling point, 1 mL of homogeneous medium was taken from each flask. The samples were processed using a rapid isolation protocol with oxalic acid, and the daunomycin concentration was measured by HPLC.

### 2.14. Changes in Media Color and Physical Parameters

The color and physical properties of media samples were assessed at 24 h intervals throughout the cultivation period, beginning after 24 h and continuing up to 264 h. At each time point, a sample was visually inspected to determine any change in color and/or hue (subjective assessment) and to observe the formation of any precipitate.

### 2.15. Observation of Morphological Changes During Fermentation

#### 2.15.1. Optical Microscopy

Samples were placed on a clean glass slide, covered with a coverslip, and examined under a Nikon Eclipse Ni-E optical microscope (Nikon Europe B.V., Amstelveen, The Netherlands) to assess morphology, size, and aggregation. Digital images were captured for further analysis and comparison over time.

#### 2.15.2. Scanning Electron Microscopy

Samples were prepared for SEM by fixation in 2.5% glutaraldehyde with phosphate buffer for 2 h at 4 °C, washing in phosphate buffer and post-fixing in 1% OsO_4_ for 1 h. The samples were dehydrated through a graded ethanol series (30%, 50%, 70%, 90%, and 100%) before mounting onto conductive stubs and sputtering with gold–palladium to enhance conductivity. Following calibration, samples were examined under high vacuum using a JSM-7401F instrument with secondary electron imaging.

#### 2.15.3. Transmission Electron Microscopy

Samples were prepared for transmission electron microscopy (TEM) by fixation in 2.5% glutaraldehyde with phosphate buffer for 2 h at 4 °C, washing in phosphate buffer and post-fixing in 1% OsO_4_ for 1 h. The samples were dehydrated through a graded ethanol series (30%, 50%, 70%, 90%, and 100%) and embedded in epoxy resin. Thin sections (~70 nm) were prepared using an ultramicrotome and placed on copper grids. The grids were stained with uranyl acetate and lead citrate to enhance contrast. The sections were then examined using a JEM-1400 instrument (JEOL Ltd., Tokyo, Japan).

### 2.16. Polyacrylamide Gel Electrophoresis and Protein Identification by MALDI-TOF

Samples were analyzed by 12% polyacrylamide gel electrophoresis (PAGE) using denaturing SDS-PAGE gels [30]. The SDS-polyacrylamide gel was stained with Coomassie Brilliant Blue, whereas proteins of interest in the native PAGE gel were naturally stained with daunomycin. Protein bands were analyzed by MALDI-TOF mass spectrometry using an Ultraflextreme instrument (Bruker Daltonik, Karlsruhe, Germany) in linear positive/negative mode with FlexControl v3.4 software. External calibration was achieved using *Escherichia coli* DH5α standard (Bruker Daltonik). The laser was set to 120% of the threshold laser power for an individual type of sample, and 1000 laser shots were acquired per sample. Mass spectra were processed using Flex Analysis v3.4 and BioTyper v3.1 (Bruker Daltonik).

### 2.17. Measurement of pH, Dissolved Oxygen, and Redox Potential During Fermentation

The pH of the fermentation media was measured using a calibrated pH meter with a glass electrode. At each sampling interval, a small volume of medium was transferred to a clean container and the reading was recorded after stabilization. The pH meter was calibrated using standard buffer solutions at pH 4.00 and 7.00. Dissolved oxygen (DO) levels were assessed using a portable meter and samples acquired at the same intervals used for pH measurements. Similarly, the redox potential was measured using a portable redox meter with a combination electrode, calibrated using standard reference solutions.

### 2.18. Extraction and Sample Preparation

For oxalic acid extraction, 400 µL of the sample was transferred to a 2 mL tube, acidified with 180 µL 1 M oxalic acid, and mixed thoroughly by vortexing. Then, 2 × 600 µL of acetone was added with further vortexing, before adding 300 µL of distilled water. After mixing, the sample was centrifuged at 14,000–15,000 rpm for 6 min, and 400 µL of the supernatant was analyzed by HPLC.

For phosphoric acid extraction, 10 mL of agitated culture broth was transferred to a 50 mL Falcon tube and mixed with 40 mL 0.1 M H_3_PO_4_. The sample was heated to 50 °C in a water bath for 30 min, with a pH check before and after heating (should be <2). We then transferred 400 µL to a 2 mL tube for analysis, and centrifuged the Falcon tube for 6–7 min at 4000–4700 rpm. After discarding the supernatant, the sediment (~5 mL) was resuspended in 40 mL 0.05 M H_3_PO_4_ and mixed until homogeneous. The pH was measured again (should be ≤1.7) before heating the sample as above with occasional mixing. The pH was measured again (should be >1.3) and a final 400 µL sample was taken for further analysis.

### 2.19. HPLC

Samples were analyzed on a ProntoSIL C18AQ column (150 × 4.6 mm, 5 µm) maintained at ambient temperature, with a flow rate of 1 mL/min and UV/Vis detection at 254 nm. The samples were separated in an isocratic 50:50 mixture of mobile phase A (1 g/L SDS in water, pH 2.7 adjusted with H_3_PO_4_) and mobile phase B (acetonitrile). The retention time of daunomycin was ~7 min in an analysis lasting ~25 min.

### 2.20. Olive Oil Extraction and HPLC

We extracted 5 g of pomace olive oil by vortexing in 2 mL methanol/water (70:30, *v*/*v*) and 2 mL hexane. Following centrifugation at 13,000 rpm for 5 min, the hydroalcoholic (settled) phase was collected for HPLC analysis on an ACE 5 μm C18 column (250 × 4.6 mm). Samples were separated in an isocratic 80:20 mixture of mobile phase A (water acidified with H_3_PO_4_, pH 3.2) and mobile phase B (acetonitrile) at 1 mL/min, with detection by UV/Vis spectroscopy at 210 nm. The retention time of daunomycin was 5.5 min. Metabolic fingerprinting was carried out by ultra-high-performance liquid chromatography coupled with tandem high-resolution mass spectrometry (U-HPLC-HRMS/MS). The samples were separated on a reversed-phase column [31] and individual compounds were detected by quadrupole/time-of-flight high-resolution mass spectrometry using a TripleTOF 6600 device (SCIEX, Toronto, ON, Canada) and PeakView v2.0 software.

## 3. Results and Discussion

### 3.1. Production Strain

The original strain showed considerable variability in mycelium, colony, and spore morphology, with dwarf spore chains formed of rudimentary spines or smooth spores as previously noted [32]. In contrast, the production strain *S. coeruleorubidus* RTA 2210 generated by repeated protoplast fusion involving triclosan-resistant strains and those capable of utilizing olive pomace oil, combined with an adaptive laboratory evolution technique, showed morphological homogeneity. The cultures underwent prolific sporulation and produced well-formed chains of spiny spores (Figure 1a,b).

When cultivated on PDA, the production strain formed homogenous colonies that secreted a red pigment (daunomycin) into the medium (Figure 2a,b).

### 3.2. Optimization of the Production Medium

#### 3.2.1. Optimization of Physical Parameters

We initially focused on the development of a stable matrix without considering production metrics, aiming to ensure the emulsification of oil while preventing the oxidation of FeSO_4_ during the autoclaving process. We then optimized production by empirically testing different combinations of components.

Transitioning from a sugar-based medium (with production of daunomycin around 2 g/L) to a more complex formulation that utilized oil as the sole carbon source posed significant challenges, particularly with the incorporation of iron as FeSO_4_. Our initial attempts to replace glucose with 10% oil led to instability, resulting in the undesirable formation of water and oil phases. Furthermore, the presence of FeSO_4_ promoted oxidation (rust formation) under our experimental conditions, revealing a lack of emulsifiers in the medium. These challenges were addressed by introducing soy derivatives, which contain natural emulsifiers rich in phospholipids and proteins that prevent phase separation by reducing surface tension at the oil–water interface, allowing the components to form beneficial complexes (Section 3.2.3).

#### 3.2.2. Replacement of Glucose with Oils

The source and quality of oil had a profound impact on the properties of the medium and the yield of daunomycin. Oils have been employed as carbon sources to produce polyketide antibiotics in *Streptomyces* and, for example, [33] reports a positive correlation between oil metabolism and salinomycin biosynthesis. Virgin olive oil was used as a control for pomace oil, whereas rapeseed oil, whose fatty acid composition closely resembles that of olive oil, was used to compare fatty acid profiles. Crude hemp oil, differing in composition but containing high concentrations of chlorophyll and phenolic compounds, was assessed for its potential influence on production (Table 1). The use of rapeseed oil reduced daunomycin yields by ~80% despite the similar fatty acid composition to olive oil. Furthermore, the use of crude hemp oil reduced yields by 95% compared to pomace oil, suggesting that chlorophyll and phenolic compounds alone do not affect daunomycin production. Notably, the use of virgin olive oil achieved only half the yield of olive pomace oil. A comparative analysis of the key differences between virgin and pomace olive oil [34] indicated that olive pomace oil is primarily composed of monounsaturated fatty acids, particularly oleic acid (C18:1), which constitutes up to 85% of the fatty acid content. Olive pomace oil also contains diverse minor components that may contribute positively to its health-promoting effects, including pentacyclic triterpenes, squalene, tocopherols, sterols, fatty alcohols, and phenolic compounds.

The use of olive pomace oil not only provides an alternative carbon source but also various natural antioxidants that mitigate the oxidative stress caused by iron. This dual functionality (nutrient supply and stress tolerance) is rarely observed in standard growth media. HPLC analysis indicated substantial changes in the composition of pomace oil over time (Figure 3a). We observed shifts across the entire chromatogram, with the most prominent characteristic peaks observed at ~5.5 min. Metabolomic analysis (Figure 3b) highlighted the differences between samples of new and old oil. In positive ionization mode, metabolomic fingerprinting by U-HPLC-HRMS/MS revealed that monoacylglycerols with acyl chain lengths of C18:1 and C18:2 were more abundant in the old oil. This probably reflects hydrolytic rancidity during storage, leading to the release of free fatty acids via the hydrolysis of ester bonds in triacylglycerols.

In positive ionization mode, higher concentrations of substances with the sum formulas C_18_H_32_O_4_ and C_22_H_40_O_8_ were detected in the old oil samples. Although these substances could not be identified, the peroxide values (PVs) of the oil samples provided important insights. The PV of the new oil was 5.87 meq O_2_/kg, whereas that of the old oil was significantly higher at 24.4 meq O_2_/kg. These elevated values suggest that the unidentified substances may be products of oxidative rancidity, specifically hydroperoxides of fatty acids. A comparison of the HPLC and U-HPLC-HRMS/MS profiles (Figure 3a,b) confirmed similar changes across the entire chromatogram, which could be used along with the PVs to assess oil quality.

Our results align with recent work reporting the ineffectiveness of expired pomace oil, confirming the key role of phenolic compounds and particularly antioxidants in microbial growth and activity [8]. Fresh pomace oil, rich in these compounds, provides antioxidants that mitigate oxidative stress [35], which is particularly important for the cultivation of sensitive strains. The degradation of phenolic compounds in expired oils suggests that factors such as air exposure, heat, and light contribute to oxidation, emphasizing the need for proper storage conditions to preserve these beneficial components. This experience highlights the need for the careful selection and monitoring of media components, especially oils, in cultivation protocols.

#### 3.2.3. Influence of Soy Source on Daunomycin Production

Three commercial soy products were tested: soy flour, defatted soy flour, and soy grits. They not only improved the physical properties of the medium and stabilized the iron, but also provided nutrients to the bacteria. All three sources were suitable as emulsifiers, although soy grits achieved ~15% higher daunomycin yields and were selected for the final cultivation medium. Because soy grits are coarse and undergo less processing than soy flour, they are likely to retain higher levels of naturally occurring phytic acid. The differences in structure and starch granule size between soy grits and soy flour may influence moisture retention, microbial accessibility, and the gradual release of nutrients over time, enhancing microbial growth. Moreover, soy grits may contain more intact proteins and fibers compared to soy flour, thus enhancing metabolic activity. Organic compounds derived from soy could act as chelators, promoting the retention of iron in its soluble ferrous state, and the phytic acid would chelate excess iron [36]. This would limit its bioavailability and minimize toxicity [37,38], which is particularly beneficial for the growth of sensitive strains. This aspect is often overlooked in standard cultivation protocols. Finally, glycinins—an important component of soy proteins—are known to play a role in iron metabolism [39]. In contrast, the high-temperature processing methods used to produce soy flour reduce the quantity of phytic acid and soluble proteins, hindering the use of soy flour in cultivation media.

#### 3.2.4. Influence of Yeast Source on Daunomycin Production

The addition of yeast to the culture medium influenced daunomycin production and pH. It is possible that maintaining a favorable pH and thereby extending the cultivation period is essentially a manifestation of the same effect. Increasing the amount of yeast from an initial dose of 1 g/L to a final dose of 10 g/L was accompanied by a stabilization of the medium’s buffering capacity and the maintenance of optimal cultivation conditions, characterized by a pH range of 6.2–7.5 for a longer duration (Table 2). The quality of the yeast is also important. Baker’s yeast achieved 20% higher yields than the same dose of brewer’s yeast (Table 3). This may reflect the more consistent quality of the baker’s yeast we used, compared to the locally sourced brewer’s yeast which can vary from batch to batch.

### 3.3. Optimal Media Composition for Daunomycin Production

We identified the optimal composition for sustained daunomycin production by empirical testing: 100 g/L pomace oil, 5.6 g/L FeSO_4_, 10 g/L yeast, 20 g/L soy grits, 5 g/L yeast extract, 5 g/L glycerol, 2 g/L K_2_HPO_4_, 3 g/L CaCO_3_, and 1 g/L MgSO_4_ (pH 6.0). The new *S. coeruleorubidus* strain RTA 2210 achieved daunomycin yields of 5.5–6.0 g/L in this medium. The repeatability of production was confirmed by the similar results of 11 replicate cultivations established by different team members (Table 4).

The extraction of daunomycin from the cultivation medium in the form of an inactive precipitate offers several advantages over the recovery of soluble daunomycin from the sugar-based medium. First, it reduces the use of chlorinated hydrocarbons as found in traditional extraction methods, thus reducing costs and environmental harm. Instead, the new process allows for the straightforward separation and extraction of the precipitate using a strong acid. Second, it enhances the recovery of daunomycin, thus improving yields. Third, the HPLC profile obtained following extraction with phosphoric acid indicates that this novel approach delivers a highly pure product, which can simplify the purification process (Figure 4). Overall, this method supports a more sustainable and cost-effective production process while upholding rigorous quality standards in drug manufacturing.

### 3.4. Changes in Media Color and Physical Parameters

The color of the medium changed during preparation and cultivation (Figure 5a). The non-sterile medium changed color from brown to olive/gray/green when autoclaved due to Maillard reactions and the degradation of certain compounds, but then remained stable without further color changes for weeks, even when stored at room temperature. During the first 48 h of cultivation, the medium changed from olive/gray/green to progressively darker gray, indicating the accumulation of metabolic byproducts. After 48 h, we observed the formation of green oily droplets and black particles (the remnants of soy grits bound to iron). The oil in the medium was not fully dispersed, creating droplets along the sides of the flask. After ~72 h, the medium turned a dark gray-black color, and a layer of oil formed on the surface, taking on a slight orange hue as the bacteria began to produce daunomycin in significant amounts (Figure 5b). During the period from 72 to 264 h, the color of the medium changed to brown-red-black as the number of visible particles at the bottom of the tilted flask increased. The oil was fully dispersed in the medium, imparting a strong orange color that indicated the presence of daunomycin. In the absence of shaking, an orange-colored oil layer containing daunomycin formed on the surface.

The initial olive/gray/green color of the autoclaved medium suggests the presence of oxidized iron in the solution (Fe^3+^) as well as other compounds, such as phytates from the soy flour or phenolic compounds from the olive pomace oil. The darkening of the medium during cultivation correlates with the more anaerobic or reducing conditions triggered by bacterial metabolism, leading to the accumulation of Fe^2+^ and the products of microbial growth. At the peak of daunomycin production, the medium becomes completely black, indicating the extensive reduction of Fe^3+^ to Fe^2+^, along with the production of complex organic molecules (including daunomycin) and their aggregation with iron. The extensive precipitation of iron complexes is also evident, due to high concentrations of organic acids and phenolic compounds.

### 3.5. The Precipitation of Iron-Containing Complexes

The black precipitate is likely to be composed of partially crystallized complexed iron (Fe^2+^ or Fe^3+^) in conjunction with organic metabolites produced during fermentation, including daunomycin. The presence of crystalline iron in the form of (Fe_3_PO_4_)_2_·8H_2_O, known as vivianite, was confirmed by XRD (Figure 6). Precipitation occurs when iron ions interact with phenolic compounds and other secondary metabolites or when pH levels shift due to microbial metabolism.

The sequestration of daunomycin within iron precipitates makes it effectively unavailable [6], mitigating the toxicity typically associated with this antibiotic and the inhibition of microbial growth and daunomycin synthesis [3]. This strategy, based on newly developed cultivation/production media, allows for the continuous production of inactivated daunomycin in the form of nanoparticles in the organo-iron complex without the adverse effects associated with the high concentrations of daunomycin freely available in the cultivation medium [2,4]. The reducing environment, characterized by low DO levels and fluctuating pH, maintains conditions that favor the precipitation of iron, facilitating the entrapment of daunomycin and enhancing the overall yield and productivity during the fermentation process. This highlights the intricate balance between microbial ecology, metabolite management, and fermentation optimization.

The detection of an amorphous organic complex signifies the formation of a non-crystalline, heterogeneous compound that may enhance the stability and bioavailability of daunomycin. The amorphous nature of the complex suggests intricate interactions take place between daunomycin and iron in the medium, highlighting the ability of the heterocyclic functional groups to chelate iron ions [40]. Chelation could alter the physicochemical properties of daunomycin, reducing its free concentration and consequently its direct cytotoxic effects. Sequestering daunomycin in this complex may effectively mitigate the acute toxic effects associated with high concentrations of the antibiotic. The stabilization provided by the amorphous organic complex may also prevent the precipitation of daunomycin, enhancing its availability for microbial uptake during the production phase.

A combination of anoxic conditions, phosphate in the medium, and bacterial activity supports the formation of vivianite during fermentation [41,42]. The black precipitate we observed suggested our fermentation system promoted the precipitation of vivianite, which was confirmed by the XRD data. This shows that iron availability is necessary to sustain microbial activity. Vivianite’s stoichiometry as a source of Fe^2+^ suggests that it could fulfill the iron requirements essential for bacterial growth while simultaneously serving as a reservoir of iron. This dual function not only supports cellular processes but also the formation of iron–daunomycin complexes, thereby reducing the bioavailability of free daunomycin and its associated toxicity. The iron-sequestering capacity of the amorphous organic complex and vivianite may help to optimize the conditions for daunomycin biosynthesis. The controlled release of iron from vivianite provides a continuous supply, ensuring that the iron is available for vital cellular functions while mitigating the risks of iron-induced oxidative stress [43].

### 3.6. Measurement of pH, DO, and Redox Potential During Fermentation

To confirm the hypothesis that color changes in the culture medium result from alterations in redox potential and iron reduction during fermentation, we measured the pH, DO, and redox potential throughout the fermentation process. By systematically correlating these data with the observed color variations, we sought to identify the most effective indicators for monitoring the cultivation process. However, we found that pH and DO were inadequate metrics for this purpose. Fluctuations in pH did not exceed 1 pH unit, and DO levels declined significantly within the first 24 h, ultimately dropping below the calibration interval of the O_2_ electrode (near-zero values). This was exacerbated by the calibration limitations of the electrode and only became evident at the point of process collapse. In contrast, the redox potential was a highly informative parameter. We observed a marked increase in redox potential immediately after autoclaving, probably reflecting the oxidation of organic compounds and the partial conversion of Fe^2+^ to Fe^3+^. This suggests that redox potential is a more useful indicator of the biochemical changes linked to color variations in the culture medium. We found an inverse relationship between redox values and daunomycin production (Figure 7). This confirms the suitability of the redox potential as a metric to optimize secondary metabolite production.

Our data suggest that changes in redox potential during cultivation can be divided into three phases: (1) before and after autoclaving, (2) during microbial growth, and (3) at the end of cultivation. These are considered in turn below.

Before autoclaving, the redox potential reflects the initial conditions of the medium, including organic substrates, DO, and metabolites produced by any contaminating microbes. The redox potential increases after autoclaving, because the degradation of organic compounds releases oxidizing agents such as hydrogen peroxide, and modifies the solubility of metal ions. Additionally, autoclaving sterilizes the medium but allows any residual DO to remain.

During the growth of *S. coeruleorubidus*, the redox potential declines in response to microbial metabolism, because the utilization of carbon sources generates reducing equivalents such as NADH. These are in turn utilized in various biochemical pathways, making the medium more reductive. The aerobic growth of the bacteria consumes DO, further promoting such conditions. This reduction process converts Fe^3+^ to Fe^2+^, making more soluble iron available for microbial growth and enzyme activity. In an Fe^2+^-enriched environment, particularly under constant aeration, the dynamic interconversion between Fe^3+^ and Fe^2+^ can significantly affect the yield of fermentation products, including daunomycin. Despite the greater availability of Fe^3+^, Fe^2+^ can still affect electron transfer paths, serving as an electron donor. Zero-valent iron, while less common in fermentation processes, can influence the overall redox balance. The accumulation of reductive metabolites, including organic acids and other reducing agents, exacerbates the decline in redox potential. Collectively, these factors highlight the dynamic interplay between microbial activity and redox potential, emphasizing how *S. coeruleorubidus* metabolic processes alter the redox environment as the culture develops.

At the end of microbial cultivation, we observed a significant increase in redox potential due to nutrient depletion, which slows down microbial metabolism. This can create oxidative conditions as the medium recovers from earlier reduced states. Additionally, cell lysis releases various intracellular compounds that can form complexes with iron and inhibit bacterial growth, further influencing the redox state. As the culture nears the end of its viability, the remaining cells may also begin to oxidize any residual substrates.

The behavior of daunomycin in the presence of iron depends on the interplay between pH and electron transfer. Fluctuating pH within a narrow range suggests buffering systems or metabolic activities affecting the medium’s acidity. The undetectable levels of DO indicate anaerobic conditions, which promote iron precipitation and favor a reducing environment. This promotes the conversion of Fe^3+^ to Fe^2+^, although the precipitation of iron oxides and hydroxides is also apparent. Daunomycin is deposited in these iron precipitates, rendering it unavailable and thereby less toxic, enhancing the productivity of the fermentation process and the overall yield.

### 3.7. Observation of Morphological Changes During Fermentation

The morphological characteristics of the particles changed during fermentation, influenced by bacterial activity, the combination of oil and iron, and other environmental conditions, resulting in three distinct types: nanoparticles, spherical particles, and large sediment particles. These forms reflect the complexity and adaptability of microbial systems in response to the environment and play a key role in the detoxification of daunomycin. The use of these mechanisms appears to be a more effective way to increase daunomycin production than the methods proposed thus far [44].

The initial stage of fermentation was marked by the rapid formation of nanoparticles ranging from 10 to 30 nm in diameter. This began shortly after inoculation, when the bacteria started to secrete compounds that facilitate the condensation of metal ions and precursors. The nanoparticles collectively have a large surface area, and detoxify the surrounding medium by binding and neutralizing daunomycin, leading to the production of stable iron-containing particles [10]. This detoxification mechanism is necessary for bacterial survival in the presence of toxins. Environmental factors such as pH, temperature, and nutrient availability affect the enzymatic activity of microbes and the kinetics of nanoparticle formation. Importantly, we found that the nanoparticles exist in two forms: attached to solid surfaces (Figure 8A) or freely floating in the medium, where they aggregate into optically dense, membrane-bound micellar structures. The lipid membrane surrounding iron–daunomycin complexes enhances their stability and acts as a further shield against toxicity, thus influencing the ecological dynamics between these *Streptomyces* and the daunomycin they produce (Figure 8G).

The aggregation of optically dense, membrane-bound micellar structures leads to the development of spherical particles, 3–5 µm in diameter (Figure 8B). These feature a shell that enhances their stability and renders them inert (Figure 8E,F). The spherical particles are eventually aggregated to form the sediment (Figure 8D).

The final stage of fermentation is characterized by the formation of large, dark sediment particles, exceeding 100 µm in diameter (Figure 8C). These particles represent a substantial portion of medium aggregates formed during fermentation and are primarily composed of dead and active mycelia, heavily coated with nanoparticles and trapped µm-sized particles. The formation of these large particles suggests the aggregation of smaller particles, facilitated by interactions with extracellular polymeric substances produced during microbial growth. Their formation may be influenced by the sedimentation of heavier complexes that arise throughout the fermentation process.

To confirm the presence of daunomycin within the large sediment particles, the protective layers were removed using hydrogen peroxide and the samples were examined under a light microscope. This resulted in the bleaching of the darker pigmentation and the subsequent release of reddish vesicles (Figure 8H). Prolonged exposure increased the number of vesicles produced. HPLC analysis of the collected vesicles confirmed the presence of daunomycin. The release of pigmented vesicles or vacuoles suggests that daunomycin is localized within membrane-bound structures inside the larger particles, rather than being present in a freely accessible form. This is likely to be a key mechanism that reduces the toxicity of daunomycin and influences the metabolic dynamics of the microbial community involved in the fermentation process. The optically dense structures identified by TEM may therefore function as reservoirs that isolate daunomycin from the surrounding environment.

The presence of iron in the particles was confirmed by a combination of SEM and EDS. However, we did not detect any L emission lines, indicating there were no significant contributions from the L shell transitions of iron. In contrast, the intensity of the K emission lines was 5.27 counts with an uncertainty of ±0.79, indicating that iron is a major component of the material (Figure 9). The Au peak in the graph corresponds to the sputtering coat applied to the sample [45].

### 3.8. Detection and Identification of Proteins

Two groups of proteins with functional similarities (managing oxidative stress and enhancing cellular resilience) were detected in the daunomycin–iron particles by SDS-PAGE followed by MALDI-TOF MS (Table 5).

In media with high iron levels, proteins in the first group such as bacterioferritin [46], catalase [47,48], oxidoreductase [49,50,51], glycinins G1 and G4, and β-conglycinin are needed to maintain iron homeostasis and mitigate oxidative stress related to elevated iron levels [39,52,53]. Bacterioferritin sequesters excess iron in a biologically accessible form and releases it when needed for cellular processes [46]. Catalase generates ROS, including hydrogen peroxide. Oxidoreductases contribute to redox balance in the presence of high iron levels by facilitating necessary metabolic reactions and detoxifying harmful compounds. Additionally, storage proteins such as glycinins G1 and G4 and β-conglycinin provide essential amino acids and may promote stress responses by modulating protein synthesis and repair. Although not quantified, phytic acid, which is present in soy grits, chelates iron, acts as an antioxidant, regulates mineral absorption, and may influence cellular signaling and gene expression. It can synergize with the aforementioned proteins, bolstering iron homeostasis and enhancing overall stress tolerance.

The second group of proteins included chemical-damaging agent resistance protein C, major outer membrane lipoprotein, xanthine dehydrogenase (YagS FAD-binding subunit), molybdopterin dehydrogenase, aklanonic acid methyl ester cyclase (DauD), a putative flavoprotein, superoxide dismutase, a nickel-containing superoxide dismutase, and a glutamate-binding protein. These proteins protect cells against chemical toxicity and oxidative stress. Chemical-damaging agent resistance protein C provides protection against harmful compounds, whereas the major outer membrane lipoprotein maintains membrane integrity under stress [54]. Xanthine dehydrogenase [55] and molybdopterin dehydrogenase [56] maintain the redox balance and counter the effects of ROS. Although aklanonic acid methyl ester cyclase is primarily associated with secondary metabolism, specifically the synthesis of anthracyclines [57], it is also essential for stress tolerance. Superoxide dismutase and nickel-containing superoxide dismutase directly combat oxidative damage, converting harmful superoxide radicals into less toxic molecules [58]. Finally, glutamate-binding protein is a member of the ABC transporter family and may promote homeostasis indirectly under stress [59]. Collectively, these proteins enhance the survival and adaptability of the production strain as part of the complex stress-response machinery.

It is important to highlight the unique origin of the *Streptomyces* strain used in this study, which was isolated from mosquito larvae. Accordingly, our work not only demonstrates a novel cultivation process for daunomycin production but also shows the potential of insect-derived microbes as valuable assets in pharmaceutical research. Insects, one of the most diverse taxa on the planet, are hosts to a vast array of microorganisms that produce a wealth of bioactive compounds relevant to drug development [60,61,62]. The metabolic diversity inherent in these insect-associated microbes, cultivated over millions of years of co-evolution with their hosts, presents a significant opportunity for advancing drug discovery, particularly in the context of antimicrobial resistance and the emergence of novel diseases [63,64]. The exploration of this underutilized microbial reservoir has the potential to uncover therapeutic agents with new mechanisms of action.

## 4. Conclusions

We have determined the mechanism of biogenic nanoparticle formation in the context of daunomycin production, an anthracycline antibiotic mainly used to treat leukemia. The inherent cytotoxicity of daunomycin and the challenges associated with industrial-scale synthesis due to microbial toxicity have limited its therapeutic applications. By developing a specialized cultivation medium that integrates olive pomace oil and iron, we have induced an autonomous resistance mechanism based on biogenic nanoparticle formation. The olive pomace oil not only serves as a carbon source, but its amphiphilic properties also facilitate the stabilization of nanoparticles, thereby enhancing the efficacy of daunomycin synthesis by optimizing the redox environment. The new medium increased daunomycin yields to 5.5–6.0 g/L (about 300% more than on sugar-based medium), a significant improvement over previous methods. The characterization of the nanoparticles confirmed the successful incorporation of iron and daunomycin, showing how this approach can mitigate cytotoxicity while improving yield. The presence of specific proteins associated with iron homeostasis and oxidative stress responses illustrates the ability of the production strain to adapt to high iron concentrations. Moreover, the inverse correlation between redox potential and daunomycin production suggests that redox monitoring could serve as a valuable indicator for optimal fermentation conditions. Our work not only contributes to the field of microbial fermentation and antibiotic production, but also emphasizes the importance of minimizing environmental impacts through the production of insoluble daunomycin precipitates that can be recovered from the cultivation medium efficiently. This precipitate is easily separated from the cultivation medium by filtration or centrifugation, concentrated into a smaller volume, and extracted using phosphoric acid, followed by a final extraction of the dissolved daunomycin using significantly reduced volumes of organic solvents, minimizing the environmental footprint of the process. Overall, our findings present promising avenues for further investigation into the mechanisms underlying biogenic nanoparticle formation and the optimization of cultivation processes. Such explorations may not only refine microbial production systems for daunomycin but also broaden the potential application of similar strategies for the synthesis of other therapeutically important compounds.

## Figures and Tables

**Figure 1 microorganisms-13-00107-f001:**
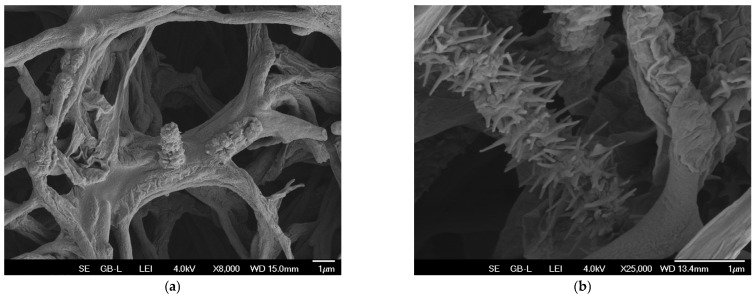
A morphological comparison of the original and production strains. (**a**) The typical morphology of the original strain characterized by underdeveloped, irregular spores forming rudimentary chains with variable morphology. (**b**) The typical morphology of the fully mature spores of the production strain generated by protoplast fusion.

**Figure 2 microorganisms-13-00107-f002:**
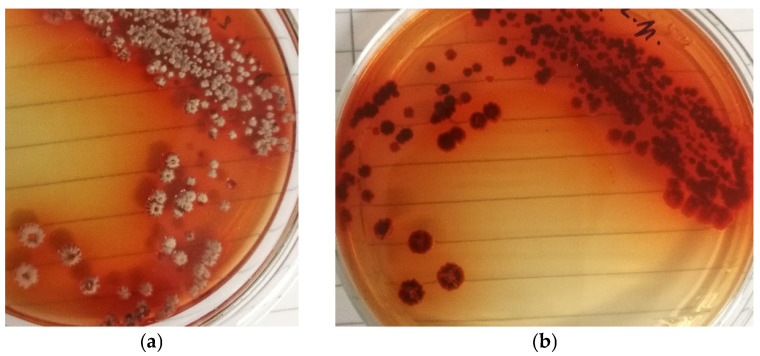
The typical behavior of production strain RTA 2210 on PDA. (**a**) Colony morphology viewed from above. (**b**) Colony morphology viewed from below, showing the secretion of a red pigment into the medium (90 mm Petri dishes).

**Figure 3 microorganisms-13-00107-f003:**
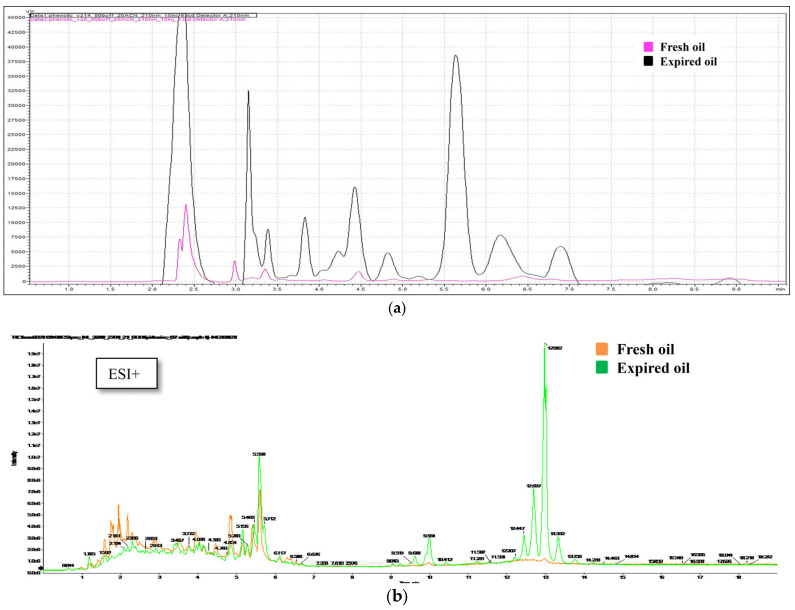
Changes in the composition of pomace oil over time. (**a**) HPLC profiles of fresh (pink) and expired (black) pomace olive oil. The red arrow indicates the most significant change. (**b**) Ion peaks representing fresh (orange) and expired (green) pomace olive oil in positive ionization mode.

**Figure 4 microorganisms-13-00107-f004:**
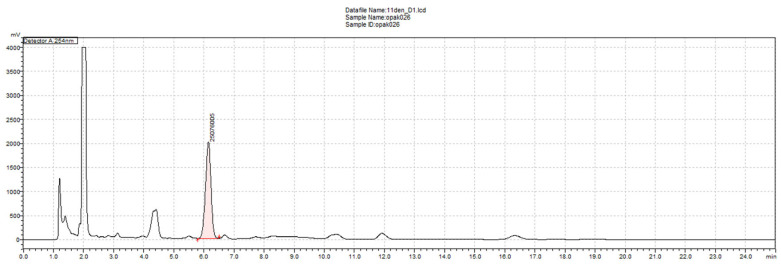
HPLC profile following daunomycin extraction from the precipitate; pink color corresponds to daunomycin peak.

**Figure 5 microorganisms-13-00107-f005:**
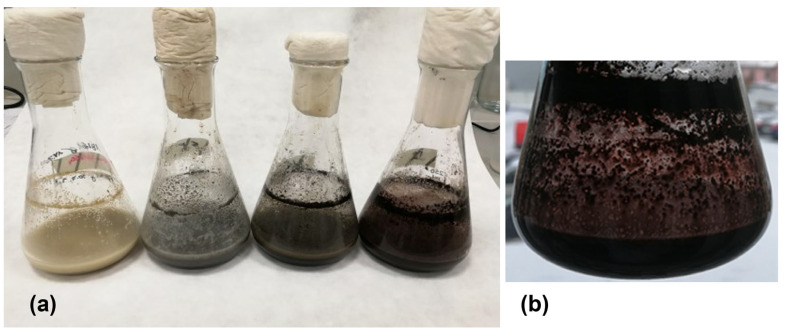
The color of the medium changes during fermentation. (**a**) The color change over a period of 246 h. (**b**) The typical appearance of the culture after 264 h.

**Figure 6 microorganisms-13-00107-f006:**
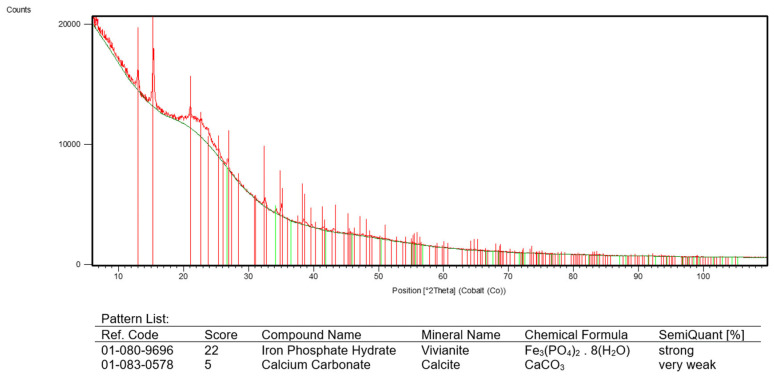
The XRD profile of the final precipitate, indicating the presence of crystalline vivianite and an amorphous organic complex.

**Figure 7 microorganisms-13-00107-f007:**
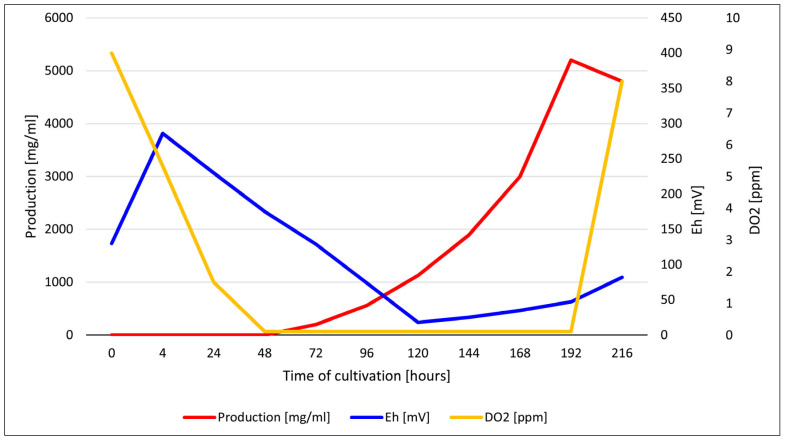
Profiles of dissolved oxygen (DO, yellow), redox potential (Eh, blue), and daunomycin production (red) during fermentation.

**Figure 8 microorganisms-13-00107-f008:**
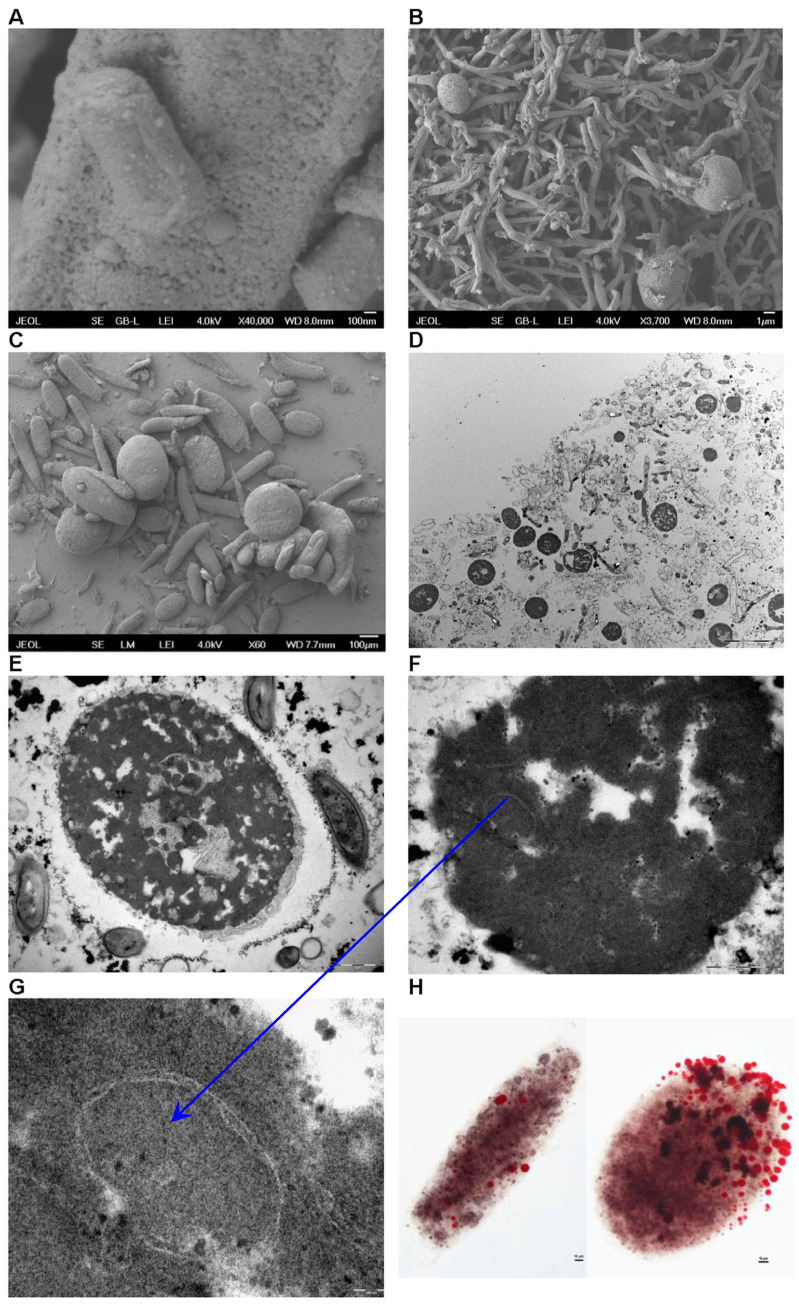
The morphological characteristics of the sediment particles. Nanoparticle interactions with mycelia: (**A**) An SEM image showing nanoparticles adhering to the mycelial surface; (**B**) an SEM image depicting spherical structures attached to a mycelium, resulting in the formation of larger sediment particles. (**C**) An SEM image showing multiple large sediment particles prominently displayed. (**D**) A TEM image showing a cross-section of a large sediment particle, highlighting spherical structures embedded within the mycelium. (**E**) A TEM image showing a cross-section of a spherical particle characterized by optically dense material. (**F**) A TEM image detailing a spherical particle, with visible membranes surrounding the optically dense structures. (**G**) A TEM image providing a closer view of the membrane encapsulating the optically dense structures that constitute the spherical particles. (**H**) Light microscope images of peroxide-treated large sediment particles. The left image shows the particle immediately following treatment, and the right image illustrates the effects of prolonged exposure to hydrogen peroxide. The red droplets represent daunomycin-containing structures released from the sediment particles.

**Figure 9 microorganisms-13-00107-f009:**
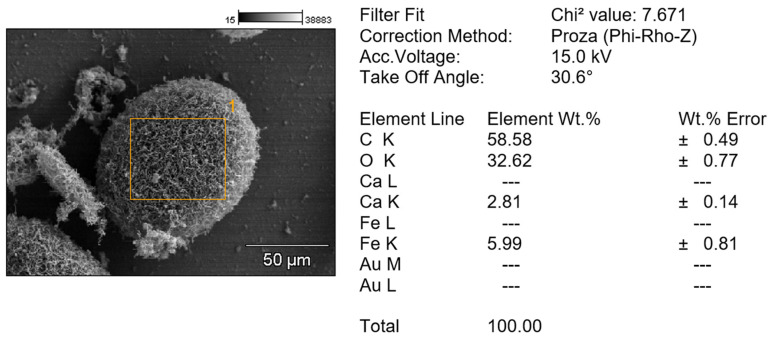
EDS detection of iron in mycelial aggregates.

**Table 1 microorganisms-13-00107-t001:** Influence of oil source on daunomycin production.

Duration of Cultivation [h]	N [Samples]	Yield [mg/L]		pH	
		Average	SD	Average	SD
Olive oil refined					
120	8	1201.3	141.7	6.03	0.03
168	8	1991.7	597.1	6.15	0.05
216	8	2220.9	492.9	8.64	0.13
264	8	2964.4	555.2	8.07	0.22
Olive oil pomace					
120	8	1710.9	86.7	6.93	0.02
168	8	2986.3	176.9	6.84	0.03
216	8	4603.7	228.9	6.87	0.03
264	8	5448.1	108.5	6.91	0.05
Rapeseed oil					
120	8	861.3	32.7	6.08	0.06
168	8	1128.5	49.3	6.14	0.05
216	8	1110.8	207.4	8.51	0.20
264	8	1089.4	88.1	7.97	0.58
Hemp oil					
120	8	61.1	17.7	6.74	0.56
168	8	125.7	11.4	7.45	0.75
216	8	168.2	42.5	8.11	0.11
264	8	152.6	41.8	8.32	0.08

**Table 2 microorganisms-13-00107-t002:** The influence of the yeast dose on the buffering capacity of the media.

Duration of Cultivation [h]	N [Samples]	Yield [mg/L]		pH	
		Average	SD	Average	SD
Baker’s yeast—1 g/L					
120	8	1116.5	656.9	7.66	0.53
168	8	1747.8	571.6	7.83	0.45
216	8	2138.6	512.5	8.47	0.27
264	8	2047.9	440.7	7.61	0.27
Baker’s yeast—5 g/L					
120	12	967.8	858.5	7.96	0.64
168	12	1877.0	862.5	7.84	0.58
216	12	2162.7	299.3	8.49	0.25
264	12	2108.6	285.8	7.81	0.32
Baker’s yeast—10 g/L					
120	8	1710.9	86.7	6.93	0.02
168	8	2986.3	176.9	6.84	0.03
216	8	4603.7	228.9	6.87	0.03
264	8	5448.1	108.5	6.91	0.05

**Table 3 microorganisms-13-00107-t003:** Influence of yeast on daunomycin production in oil-based media.

Duration of Cultivation [h]	N [Samples]	Yield [mg/L]		pH	
		Average	SD	Average	SD
Baker’s yeast—10 g/L					
120	8	1710.9	86.7	6.93	0.02
168	8	2986.3	176.9	6.84	0.03
216	8	4603.7	228.9	6.87	0.03
264	8	5448.1	108.5	6.91	0.05
Brewer’s yeast—10 g/L					
120	7	600.6	154.7	6.87	0.04
168	7	1141.2	293.2	6.78	0.03
216	7	3353.8	279.8	6.70	0.02
264	7	4240.1	325.1	6.75	0.03

**Table 4 microorganisms-13-00107-t004:** Production of daunomycin in optimized media.

Duration of Cultivation [h]	N [Samples]	Yield [mg/L]		pH	
		Average	SD	Average	SD
120	11	1389.0	350.7	6.40	0.17
168	11	3070.2	724.6	6.62	0.13
216	11	4728.2	726.1	6.72	0.10
264	11	5466.4	681.4	7.24	0.41

**Table 5 microorganisms-13-00107-t005:** Proteins detected in daunomycin–iron particles.

Group One:	Group Two:
Bacterioferritin Catalase Oxidoreductase Glycinin G1 Glycinin G4 β-Conglycinin	Chemical-damaging agent resistance protein C Major outer membrane lipoprotein Xanthine dehydrogenase Molybdopterin dehydrogenase Aklanon acid methyl ester cyclase (DauD) Superoxide dismutase Nickel-containing superoxide dismutase Glutamate-binding protein

## Data Availability

The original contributions presented in this study are included in the article. Further inquiries can be directed to the corresponding author.

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
