# Peer review of "Autonomous Defense Based on Biogenic Nanoparticle Formation in Daunomycin-Producing Streptomyces"

_microorganisms, 2025, doi:10.3390/microorganisms13010107_

Round 1

Reviewer 1 Report

Comments and Suggestions for Authors

Dear authors:

I believe that this work is comprehensive and contributes positively in research on microbial production systems for the synthesis of highly relevant therapeutic agents. However, there are some considerations that should be taken into account, which you can find in the attached file.

Regards

Author Response

Thank you very much for your time in reviewing this manuscript and for your helpful comments and questions. Detailed responses to questions/comments are provided in the attached file and the relevant revisions/corrections are included in the revised manuscript - in the track-changes mode.

Reviewer 2 Report

Comments and Suggestions for Authors

In their manuscript entitled “Autonomous Defense based on Biogenic Nanoparticle Formation in Daunomycin-Producing Streptomyces” the authors describe an empirical fermentation process carried out in presence of oil and high iron concentration that promotes the formation of particles containing daunomycin, iron as well as various proteins involved in the resistance to oxidative stress that might result from high iron availability. The formation of daunomycin-iron chelates likely protects the bacteria from the toxicity of daunomycin and thus leads to an increase of daunomycin production yields.
The paper is interesting but is too long and rather difficult to follow since it involves a lot of comments/speculations to tentatively explain the obtained results so separating the result and discussion sections may improve it readability.

The manuscript starts with a rather short introduction. The authors should mention in the introduction what are the other identified resistance mechanisms against daunomycin and if the genes encoding these other resistance mechanisms are or not linked to the biosynthetic pathway? It would also be nice to know what is the impact of low/high phosphate and nitrogen availability on daunomycin production.

Results and discussion are grouped in an unique section 3 which is not, in my opinion, the best organization for clarity. This section is divided in 8 sub-sections.

3.1 explains how the production strain S. coeruleorubidus RTA 2210 was obtained by protoplasts fusion of triclosan-resistant strains with strains able to utilize olive pomace oil. The triclosan induced mutations are supposed to impair the ability of the strain to synthetize fatty acids. Is it the case?  What are the consequence of these mutations of the ability of the strain to use external oil? What was the rational to use this strategy? In my opinion Figures 1a and 1b should be shown at the same magnification. In Figure 2, what is the red pigment secreted by the strain, is it daunomycin ?

3.2 deals with optimization of the production medium that requires good oil emulsification and protection of FeSO4 from oxidation. This was achieved by the introduction of soy derivatives that act as natural emulsifiers and nutrients source as well as by olive pomace oil that likely brings components with anti-oxidant properties. Both soy and pomace oil components might limit iron bioavailability and thus oxidative stress caused by it. In figure 3a the authors should use more contrasted colors for the two oils (new and old) since black and blue lines seems alike. The huge differences between these two chromatograms are really surprising? Was it exactly the amount of oil analyzed for the two oils? It would be nice if the authors could name on the chromatogram the identity of the chemical species identified. Furthermore the authors mentioned to use yeast to stabilize medium’s buffering capacity? Is it yeast or yeast extracts? Are the authors conducting a co-culture yeast/Streptomyces or is yeast simply a nutrient/ amino acids/vitamins source? Please clarify this point. Why should yeast stabilize the pH of the medium?

The authors should propose a section 4 (rather than 3.2.5) entitled “Cultivation in optimized medium”, once they have defined the optimized medium and are carrying out cultivation with it. In my opinion, lines 417 to 426 should be part of a separate Discussion section.

4.1 Changes in media color and physical parameters: was a lambda scan achieved on the growth medium to assess more precisely color changes mentioned by the authors?

4.2 The precipitation of iron-containing complexes: Since vivianite is mentioned in this section text of lines 491 to 503 should be transferred in this section and the subsection entitled “The presence of vivianite” should be canceled. Lines 470 to 478 could constitute a part of a separate discussion section

4.3 Measurement of pH, DO and redox potential during fermentation: In figure 7, the authors should provide the growth curve too, if possible (I am aware that the medium is complex and that biomass weight cannot be used ). Why does DO goes up at late time points ?
Daunomycin, as actinorhodin (ACT), is a polyketide. It bears quinone groups and is thus a redox active compound that can act as an electrons shuttle. ACT was shown to bear anti-oxidant (doi: 10.1038/s41598-020-65087-w.) as well as anti-respiratory properties (doi: 10.1038/s41598-017-00259-9) and its biosynthesis is likely to be induced by oxidative stress (doi: 10.3389/fmicb.2021.813993. eCollection 2021). The measure of the redox potential might be considered as a direct measure of Oxidative Stress since it indicates the relative proportions of oxidants (ROS) to reductants (antioxidants).   So low redox potential might indicate high oxidative stress (OxS) and OxS was proposed to be an important trigger of the biosynthesis of many polyketides including ACT (doi: 10.3390/antibiotics9020083). The authors should comment of these overlooked aspects and should assess whether oxidative stress is an important trigger of Daunomycin biosynthesis.

4.4 Sediment friability: The assessment of the friability of the sediment is, in my opinion not very important. So, the manuscript being already very long, this part should be either canceled or transferred in supplementary data. Furthermore no color code is given in Figure 8 about this matter. What is important is the composition of the sediment.

4.4 Progressive formation of sediment particles during fermentation: What do the authors mean by precursors line 590? Does the lipid membrane surrounding iron-daunomycin complexes originate from olive oil present in the medium of from the bacteria (line 598)? Line 605 is “aggregated” not a better term than “trapped”? It would be nice if the authors organized their pictures of Figure 9 in a temporal order and if they summarize the content of their pictures by providing annotated drawings that show the various steps of sediment particles formation throughout the cultivation process or at least the authors should name the structures shown in the pictures. The round and elongated structures of various size seen in Figure 9C are they all sediment particles? Where is the membrane in Figure 9F? Figure 9 F should be canceled in my opinion.

The authors should perhaps also mention some papers that are related to their work: DOI: 10.1111/j.1365-2672.2008.03975.x - DOI: 10.7161/omuanajas.287668 - DOI: 10.1128/AEM.00763-20

Author Response

(The authors gave the same response as above.)

Reviewer 3 Report

Comments and Suggestions for Authors

The manuscript mainly explores how autonomous defense mechanisms can be achieved through the formation of bio-nanoparticles in Streptomyces that produce daunomycin, a chemotherapy drug used to treat leukemia. The study aims to address the cytotoxicity of daunomycin, improve its yield, and explore its potential for industrial-scale synthesis. The originality of this study lies in the development of a special culture medium that induces the formation of bio-nanoparticles by daunomycin-producing Streptomyces by using oil instead of sugar as the main carbon source, thereby increasing the yield of daunomycin and potentially alleviating its cytotoxicity. This study fills the research gap in optimizing the production process of daunomycin to reduce cytotoxicity and increase yield. This study provides a new strategy to increase the yield of daunomycin and potentially reduce its toxicity by using a combination of oil-based culture medium and iron.

The conclusions in the manuscript are generally consistent with the experimental data and arguments provided. The authors experimentally demonstrated how oil-based culture medium can improve the yield of daunomycin and explored the mechanism of bio-nanoparticle formation. The references in the manuscript seem appropriate, covering previous studies in related fields, including the biosynthesis of daunomycin, the culture techniques of Streptomyces, and the formation of bionanoparticles.

Major concerns

1. The mechanism of bionanoparticle formation is not mentioned, and the authors should at least discuss it in the Discussion section.

2. How nanoparticles affect the bioavailability and toxicity of daunomycin is an unavoidable issue and needs to be discussed directly.

3. The specific method of mass spectrometry detection should be clearly described.

Minor bugs

1. "Daunomycin produced in the oilbased medium was predominantly found in the solid sediment, whereas that produced in the sugar based medium was mostly soluble." Oilbased and sugar based should be Oil-based and sugar-based.

2. "Rather than relying on the genetic modification of the production strain, we instead induced the physiological adaptation of the strain to its toxic metabolites." The readability is poor and the semantics are unclear. Please rewrite.

3. There are too many problems with the references. a. https://https:// appears multiple times; b. some species names are not italicized; c. the format is not uniform; d. some information is missing.

Author Response

(The authors gave the same response as above.)
